# Effect of top-down connections in Hierarchical Sparse Coding

## Abstract

Hierarchical Sparse Coding (HSC) is a powerful model to efficiently represent multi-dimensional, structured data such as images. The simplest solution to solve this computationally hard problem is to decompose it into independent layerwise subproblems. However, neuroscientific evidence would suggest inter-connecting these subproblems as in the Predictive Coding (PC) theory, which adds top-down connections between consecutive layers. In this study, a new model called 2-Layers Sparse Predictive Coding (2L-SPC) is introduced to assess the impact of this inter-layer feedback connection. In particular, the 2L-SPC is compared with a Hierarchical Lasso (Hi-La) network made out of a sequence of Lasso layers. 2L-SPC and a 2-layers Hi-La networks are trained on 4 different databases and with different sparsity parameters on each layer. First, we show that the overall prediction error generated by 2L-SPC is lower thanks to the feedback mechanism as it transfers prediction error between layers. Second, we demonstrate that the inference stage of the 2L-SPC is faster to converge than for the Hi-La model. Third, we show that the 2L-SPC also accelerates the learning process. Finally, the qualitative analysis of both models dictionaries, supported by their activation probability, show that the 2L-SPC features are more generic and informative.

## 1 Introduction

Finding a "efficient" representation to model a given signal in a concise and efficient manner is an inverse problem that has always been central to the machine learning community. Sparse Coding (SC) has proven to be one of the most successful methods to achieve this goal. SC holds the idea that signals (e.g. images) can be encoded as a linear combination of few features (called atoms) drawn from a bigger set called the dictionary (Elad, 2010). The pursuit of optimal coding is usually decomposed into two complementary subproblems: inference (coding) and dictionary learning. Inference consists in finding an accurate sparse representation of the input data considering the dictionaries are fixed, it could be performed using algorithms like ISTA & FISTA (Beck & Teboulle, 2009), Matching Pursuit (Mallat & Zhang, 1993), Coordinate Descent (Li & Osher, 2009), or ADMM (Heide et al., 2015). Once the representation is inferred, one can learn the atoms from the data using methods like gradient descent (Rubinstein et al., 2010; Kreutz-Delgado et al., 2003; Sulam et al., 2018), or online dictionary learning (Mairal et al., 2009a). Consequently, SC offers an unsupervised framework to learn simultaneously basis vectors (e.g. atoms) and the corresponding input representation. SC has been applied with success to image restoration (Mairal et al., 2009b), feature extraction (Szlam et al., 2010) and classification (Yang et al., 2011; Perrinet & Bednar, 2015). Interestingly, SC is also a field of interest for computational neuroscientists. Olshausen & Field (1997) first demonstrated that adding a sparse prior to a shallow neural network was sufficient to account for the emergence of neurons whose Receptive Fields (RFs) are spatially localized, band-pass and oriented filters, analogous to those found in the primary visual cortex (V1) of mammals (Hubel & Wiesel, 1962). Because most of the SC algorithms are limited to single-layer network, they cannot model the hierarchical structure of the visual cortex. However, few solutions have been proposed to tackle Hierarchical Sparse Coding (HSC) as a global optimization problem (Sulam et al., 2018; Makhzani & Frey, 2013; 2015; Aberdam et al., 2019; Sulam et al., 2019). These methods are looking for an optimal solution of HSC without considering their plausibility in term of neuronal implementation. Consequently, the quest for reliable HSC formulation that is compatible with a neural implementation remains open.

Rao & Ballard (1999) introduce the Predictive Coding (PC) to model the effect of the interaction of cortical areas in the visual cortex. PC intends to solve the inverse problem of vision by combining feedforward and feedback connections. In PC, feedback connection carries prediction of the neural activity of the lower cortical area while feedforward pass prediction error to the higher cortical area. In such a framework, neural population are updated to minimize the unexpected component of the neural signal (Friston, 2010). PC has been applied for supervised object recognition (Wen et al., 2018; Han et al., 2018; Spratling, 2017) or unsupervised prediction of future video frames (Lotter et al., 2016). Interestingly, PC is flexible enough to introduce a sparse prior to each layer. Therefore, one can consider PC as a bio-plausible formulation of the HSC problem. This formulation is to confront with the other bio-plausible HSC formulation that consists of a stack of independent Lasso problems (Sun et al., 2017). To the best of our knowledge, no study has compared these two mathematically different formulations of the same problem of optimizing the Hierarchical Sparse Coding of images. What is the effect of top-down connection of PC? What are the consequences in term of computations and convergence? What are the qualitative differences concerning the learned atoms?

The objective of this study is to experimentally answer these questions and to show that the PC framework could be successfully used for improving solutions to HSC problems. We start our study by defining the two different mathematical formulations to solve the HSC problem: the Hierarchical Lasso (Hi-La) that consists in stacking Lasso sub-problems, and the 2-Layers Sparse Predictive Coding (2L-SPC) that leverages PC into a deep and sparse network of bi-directionally connected layers. To experimentally compare both models, we train the 2L-SPC and Hi-La networks on 4 different databases and we vary the sparsity of each layer. First, we compare the overall prediction error of the two models and we break it down to understand its distribution among layers. Second, we analyze the number of iterations needed for the state variables of each network to reach their stability. Third, we compare the convergence of both models during the dictionary learning stage. Finally, we discuss the qualitative differences between the features learned by both networks in light of their activation probability.

## 2 METHODS

### 2.1 BACKGROUND

In our mathematical description, italic letters are used as symbols for *scalars*, bold lower case letters for column $\boldsymbol{vectors}$, bold uppercase letters for **MATRICES** and $\nabla_x \mathcal{L}$ denotes the gradient of $\mathcal{L}$ w.r.t. to $x$. The core objective of Hierarchical Sparse Coding (HSC) is to infer the internal state variables $\{\boldsymbol{\gamma}_i^{(k)}\}_{i=1}^L$ (also called sparse map) for each input image $\boldsymbol{x}^{(k)}$ and to learn the parameters $\{\mathbf{D}_i\}_{i=1}^L$ that solved the inverse problem formulated in Eq. 1.

$$\begin{cases} \boldsymbol{x}^{(k)} = \mathbf{D}_1^T \boldsymbol{\gamma}_1^{(k)} + \boldsymbol{\epsilon}_1^{(k)} & \text{s.t. } \|\boldsymbol{\gamma}_1^{(k)}\|_0 < \alpha_1 \quad \text{and} \quad \boldsymbol{\gamma}_1^{(k)} > 0 \\ \boldsymbol{\gamma}_1^{(k)} = \mathbf{D}_2^T \boldsymbol{\gamma}_2^{(k)} + \boldsymbol{\epsilon}_2^{(k)} & \text{s.t. } \|\boldsymbol{\gamma}_2^{(k)}\|_0 < \alpha_2 \quad \text{and} \quad \boldsymbol{\gamma}_2^{(k)} > 0 \\ .. \\ \boldsymbol{\gamma}_{L-1}^{(k)} = \mathbf{D}_L^T \boldsymbol{\gamma}_L^{(k)} + \boldsymbol{\epsilon}_L^{(k)} & \text{s.t. } \|\boldsymbol{\gamma}_L^{(k)}\|_0 < \alpha_L \quad \text{and} \quad \boldsymbol{\gamma}_L^{(k)} > 0 \end{cases} \quad (1)$$

$L$ is the number of layers and $\boldsymbol{\epsilon}_i^{(k)}$ is the layer $i$ prediction-error corresponding to the input $\boldsymbol{x}^{(k)}$. The map $\boldsymbol{\gamma}_i^{(k)}$ could be viewed as the projection of $\boldsymbol{\gamma}_{i-1}^{(k)}$ in the basis described by the atoms (denoted also basis vector) composing $\mathbf{D}_i$. $\boldsymbol{\epsilon}_i^{(k)}$ is historically called 'prediction error' but it is actually quantifying the local reconstruction error between $\mathbf{D}_i^T \boldsymbol{\gamma}_i^{(k)}$ and $\boldsymbol{\gamma}_{i-1}^{(k)}$. The sparsity of the internal state variables, specified by the $\ell_0$ pseudo-norm, is constrained by the scalar $\alpha_i$. In practice we use 4-dimensional tensors to represent both vectors and matrices. $\boldsymbol{x}^{(k)}$ is a tensor of size $[1, c_x, w_x, h_x]$ with $c_x$ being the number of channels of the image, $w_x$ and $h_x$ the width and height of the image respectively. In our mathematical description we raveled $\boldsymbol{x}^{(k)}$ as a vector of size $[c_x \times w_x \times h_x]$. Furthermore, we impose a 2-dimensional convolutional structure to the parameters $\{\mathbf{D}_i\}_{i=1}^L$. If one describes the dictionary as a 4-dimensional tensor of size $[n_d, c_d, w_d, h_d]$, one can derive $\mathbf{D}_i$ as a matrix of $n_d$ local features (size: $c_d \times w_d \times h_d$) that cover every possible location of the input $\boldsymbol{x}^{(k)}$ (Sulam et al., 2018). In other words, $\mathbf{D}_i$ is a Toeplitz matrix. For the sake of concision in our mathematical descriptions, we use matrix/vector multiplication in place of convolution as it is mathematically strictly equivalent. Replaced in a biological context $\mathbf{D}_i$ could be interpreted as the synaptic weights between two neural populations whose activity is represented by $\boldsymbol{\gamma}_{i-1}$ and $\boldsymbol{\gamma}_i$ respectively.

## 2.2 FROM HIERARCHICAL LASSO ...

One possibility to solve Eq. 1 while keeping the locality of the processing required by neural implementation, is to minimize a loss for each layer corresponding to the addition of the squared $\ell_2$-norm of the prediction error with a sparsity penalty. To obtain a convex cost, we relax the $\ell_0$ constraint into a $\ell_1$-penalty. It defines, therefore, a loss function for each layer in the form of a standard Lasso problem (Eq. 2), that could be minimized using gradient-based methods:

$$\mathcal{F}\big(\mathbf{D}_i, \boldsymbol{\gamma}_i^{(k)}\big) = \frac{1}{2}\|\boldsymbol{\gamma}_{i-1}^{(k)} - \mathbf{D}_i^T\boldsymbol{\gamma}_i^{(k)}\|_2^2 + \lambda_i\|\boldsymbol{\gamma}_i^{(k)}\|_1 \tag{2}$$

In particular, we use the Iterative Shrinkage Thresholding Algorithm (ISTA) to minimize $\mathcal{F}$ w.r.t. $\boldsymbol{\gamma}_i^{(k)}$ (Eq. 3) as it is proven to be computationally cheap (Beck & Teboulle, 2009). In practice, we use an accelerated version of the ISTA algorithm called FISTA. In a convolutional case in which the proximal operator has a closed-form, FISTA has the advantage to converge faster than other sparse coding algorithms (e.g. Coordinate Descent) (Chalasani et al., 2013). Note that in Eq. 3 we have removed image indexation to keep a concise notation.

$$\boldsymbol{\gamma}_i^{t+1} = \mathcal{T}_{\eta_{c_i}\lambda_i}\big(\boldsymbol{\gamma}_i^t - \eta_{c_i}\nabla_{\boldsymbol{\gamma}_i^t}\mathcal{F}\big) = \mathcal{T}_{\eta_{c_i}\lambda_i}\big(\boldsymbol{\gamma}_i^t + \eta_{c_i}\mathbf{D}_i(\boldsymbol{\gamma}_{i-1}^t - \mathbf{D}_i^T\boldsymbol{\gamma}_i^t)\big) \tag{3}$$

In Eq. 3, $\mathcal{T}_\alpha(\cdot)$ denotes the non-negative soft-thresholding operator, $\eta_{c_i}$ is the learning rate of the inference process and $\boldsymbol{\gamma}_i^t$ is the state variable $\boldsymbol{\gamma}_i$ at time $t$. Interestingly, one can interpret Eq. 3 as one loop of a recurrent layer that we will call the Lasso layer (Gregor & LeCun, 2010). Following Eq. 3, $\mathbf{D}_i^T$ is a decoding dictionary that back-projects $\boldsymbol{\gamma}_i$ into the space of the $(i-1)$-th layer. This back-projection is used to elicit an error with $\boldsymbol{\gamma}_{i-1}$ that will be encoded by $\mathbf{D}_i$ to update the state variables $\boldsymbol{\gamma}_i$. Finally, Lasso layers can be stacked together to form a Hierarchical Lasso (Hi-La) network (see Fig. 1 without blue arrow). The inference of the overall Hi-La network consists in updating recursively all the sparse maps until they have reached a stable point.

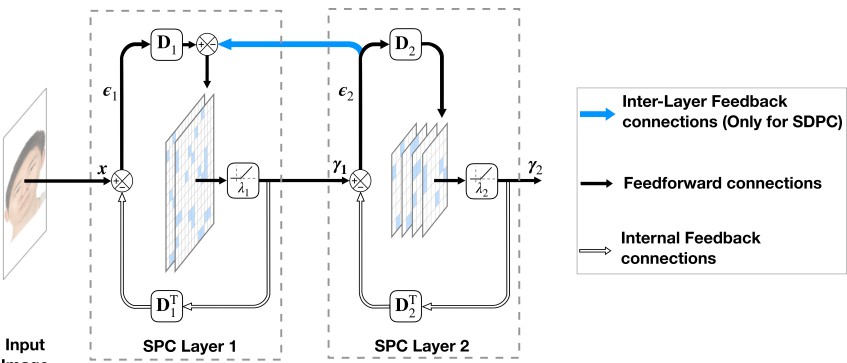

Figure 1: Inference update scheme for the 2L-SPC network. $\boldsymbol{x}$ is the input image, and $\lambda_i$ tunes the sparseness level of $\boldsymbol{\gamma}_i$. The encoding and decoding dictionaries ($\mathbf{D}_i$ and $\mathbf{D}_i^T$, respectively) are reciprocal. The sparse maps ($\boldsymbol{\gamma}_i$) are updated through a bi-directional dynamical process (plain and empty arrows). This recursive process alone describes the Hi-La network. If we add the top-down influence, called inter-layer feedback connection (blue arrow), it then becomes a 2L-SPC network.

## 2.3 ... TO HIERARCHICAL PREDICTIVE CODING

Another alternative to solve Eq. 1 is to use the Predictive Coding (PC) theory. Unlike the Lasso loss function, PC is not only minimizing the bottom-up prediction error, but it also adds a top-down prediction error that takes into consideration the influence of the upper-layer on the current layer (see Eq. 4). In other words, finding the $\boldsymbol{\gamma}_i$ that minimizes $\mathcal{L}$ consists in finding a trade-off between a representation that best predicts the lower level activity and another one that is best predicted by the upper-layer.

$$\mathcal{L}\big(\mathbf{D}_i, \boldsymbol{\gamma}_i^{(k)}\big) = \mathcal{F}\big(\mathbf{D}_i, \boldsymbol{\gamma}_i^{(k)}\big) + \frac{1}{2}\|\boldsymbol{\gamma}_i^{(k)} - \mathbf{D}_{i+1}^T\boldsymbol{\gamma}_{i+1}^{(k)}\|_2^2 \tag{4}$$

For consistency, we also use the ISTA algorithm to minimize $\mathcal{L}$ w.r.t $\boldsymbol{\gamma}_i$. The update scheme is described in Eq. 5 (without image indexation for concision):

$$\boldsymbol{\gamma}_i^{t+1} = \mathcal{T}_{\eta_{c_i} \lambda_i} \big( \boldsymbol{\gamma}_i^t - \eta_{c_i} \nabla_{\boldsymbol{\gamma}_i^t} \mathcal{L} \big) = \mathcal{T}_{\eta_{c_i} \lambda_i} \big( \boldsymbol{\gamma}_i^t + \eta_{c_i} \mathbf{D}_i (\boldsymbol{\gamma}_{i-1}^t - \mathbf{D}_i^T \boldsymbol{\gamma}_i^t) - \eta_{c_i} (\boldsymbol{\gamma}_i^t - \mathbf{D}_{i+1}^T \boldsymbol{\gamma}_{i+1}^t) \big) \quad (5)$$

Fig. 1 shows how we can interpret this update scheme as a loop of a recurrent layer. This recurrent layer, called Sparse Predictive Coding (SPC) layer, forms the building block of the 2-Layers Sparse Predictive Coding (2L-SPC) network (see Algorithm 2 in Appendix for the detailed implementation of the 2L-SPC inference). The only difference with the Hi-La architecture is that the 2L-SPC includes an inter-layer feedback connection to materialize the influence coming from upper-layers (see the blue arrow in Fig. 1).

## 2.4 Coding stopping criterion and unsupervised learning

For both networks, the inference process is finalized once the relative variation of $\boldsymbol{\gamma}_i^t$ w.r.t to $\boldsymbol{\gamma}_i^{t-1}$ is below a threshold denoted $T_{stab}$. In practice, the number of iterations needed to reach the stopping criterion is between 30 to 100 (see Fig.4 for more details). Once the convergence is achieved, we update the dictionaries using gradient descent (see Algorithm. 1). It was demonstrated by Sulam et al. (2018) that this alternation of inference and learning offers reasonable convergence guarantee. The learning of both Hi-La and 2L-SPC consists in minimizing the problem defined in Eq. 6 in which N is the number of images in the dataset. The learning occurs during the training phase only. Conversely, the inference process is the same during both training and testing phases.

$$\min_{\{\mathbf{D}_i\}} \big( \frac{1}{N} \sum_{k=1}^{N} \sum_{i=1}^{L} \mathcal{F}(\mathbf{D}_i, \boldsymbol{\gamma}_i^{(k)}) \big) \quad (6)$$

For both models, dictionaries are randomly initialized using the standard normal distribution (mean 0 and variance 1) and all the sparse maps are initialized to zero at the beginning of the inference process. After every dictionary update, we $\ell_2$-normalize each atom of the dictionary to avoid any redundant solution. Interestingly, although the inference update scheme is different for the two models, the dictionary learning loss is the same in both cases since the top-down prediction error term in $\mathcal{L}$ does not depend on $\boldsymbol{D}_i$ (see Eq. 6). This loss is then a good evaluation point to assess the impact of both 2L-SPC and Hi-La inference process on the layer prediction error $\boldsymbol{\epsilon}_i$. We used PyTorch 1.0 to implement, train, and test all the models described above. The code of the two models and the simulations of this paper are available at www.github.com/XXX/XXX.

---

**Algorithm 1:** Alternation of inference and learning for training and testing

**for** $\boldsymbol{x}^{(k)}$ in the training set **do**
     $\forall i, \boldsymbol{\gamma}_i^{(k)} = \mathbf{0}$     # initialization
     **while** *convergence not reached* **do**
         **for** $i = 1$ **to** $L$ **do**
             $\boldsymbol{\gamma}_i^{(k)} \leftarrow \mathcal{T}_{\eta_{c_i} \lambda_i} \big( \boldsymbol{\gamma}_i^{(k)} - \eta_{c_i} \nabla_{\boldsymbol{\gamma}_i^{(k)}} \mathcal{L} \big)$     # inference
     **for** $i = 1$ **to** $L$ **do**
         $\mathbf{D}_i \leftarrow \mathbf{D}_i - \eta_i \nabla_{\boldsymbol{D}_i} \mathcal{L}$       # learning (only for training)

---

## 3 Experimental settings: datasets and parameters

We use 4 different databases to train and test both networks.

**STL-10**. The STL-10 database (Coates et al., 2011) is made of 100000 colored images of size $96 \times 96$ pixels (px) representing 10 classes of objects (airplane, bird...). STL-10 presents a high diversity of objects view-points and background. This set is partitioned into a training set composed of 90000 images, and a testing set of 10000 images.

**AT&T**. The AT&T database (ATT, 1994) is made of 400 grayscale images of size $92 \times 112$ pixels (px) representing faces of 40 distinct subjects with different lighting conditions, facial expressions, and details. This set is partitioned into batches of 20 images. The training set composed of 330 images (33 subjects) and the testing set id composed of 70 images (7 subjects).

**CFD**. The Chicago Face Database (CFD) (Ma et al., 2015) consists of $1,804$ high-resolution $(2,444 \times 1,718$ px), color, standardized photographs of male and female faces of varying ethnicity between the ages of 18 and 40 years. We re-sized the pictures to $170 \times 120$ px to keep reasonable computational time. The CFD database is partitioned into batches of 10 images. This dataset is split into a training set composed of 721 images and a testing set of 486 images.

**MNIST**. MNIST (LeCun, 1998) is composed of $28 \times 28$ px, $70,000$ grayscale images representing handwritten digits. We decomposed this dataset into batches of 32 images. This dataset is split into a training set composed of $60,000$ digits and a testing set of $10,000$ digits.

All these databases are pre-processed using Local Contrast Normalization (LCN) and whitening. LCN is inspired by neuroscience and consists in a local subtractive and divisive normalization (Jarrett et al., 2009). In addition, we use whitening to reduce dependency between pixels.

To draw a fair comparison between the 2L-SPC and Hi-La models, we train both models using the same set of parameters. All these parameters are summarized in Table 1 for STL-10, MNIST and CFD databases and in Appendix C for ATT database. Note that the parameter $\eta_{c_i}$ is omitted in the table because it is computed as the inverse of the largest eigenvalue of $\mathbf{D}_i^T \mathbf{D}_i$ (Beck & Teboulle, 2009). To learn the dictionary $\mathbf{D}_i$, we use stochastic gradient descent on the training set only, with a learning rate $\eta_{L_i}$ and a momentum equal to $0.9$. In this study, we consider only 2-layered networks and we vary the sparsity parameters of each layer ($\lambda_1$ and $\lambda_2$) to assess their effect on both 2L-SPC and Hi-La networks.

Table 1: Network architectures, training and simulation parameters. The size of the convolutional kernels are shown in the format: [# features, # channels, width, height] (stride). To describe the range of explored parameters during simulations, we use the format $[0.3 : 0.7 :: 0.1, 0.5]$ ,which means that we vary $\lambda_1$ from 0.3 to 0.7 by step of 0.1 while $\lambda_2$ is fixed to 0.5.

| | | DataBases | | |
| | | STL-10 | CFD | MNIST |
|---|---|---|---|---|
| network param. | $\mathbf{D}_1$ size | [64, 1, 8, 8] (2) | [64, 3, 9, 9] (3) | [32, 1, 5, 5] (2) |
| | $\mathbf{D}_2$ size | [128, 64, 8, 8] (1) | [128, 64, 9, 9] (1) | [64, 32, 5, 5] (1) |
| | $T_{stab}$ | 1e-4 | 5e-3 | 5e-4 |
| training param. | # epochs | 10 | 250 | 100 |
| | $\eta_{L_1}$ | 1e-4 | 1e-4 | 5e-2 |
| | $\eta_{L_2}$ | 5e-3 | 5e-3 | 1e-3 |
| simu. param. | $\lambda_1$ range | $[0.2 : 0.6 :: 0.1, 1.6]$ | $[0.3 : 0.7 :: 0.1, 1.8]$ | $[0.1 : 0.3 :: 0.05, 0.3]$ |
| | $\lambda_2$ range | $[0.4, 1.4 : 1.8 :: 0.1]$ | $[0.5, 1 : 1.8 :: 0.2]$ | $[0.2, 0.2 : 0.4 :: 0.05]$ |

## 4 RESULTS

For cross-validation, we run 7 times all the simulations presented in this section, each time with a different random dictionary initialization. We define the central tendency of our curves by the median of the runs, and its variation by the Median Absolute Deviation (MAD) (Pham-Gia & Hung, 2001). We prefer this measure to the classical mean $\pm$ standard deviation because a few measures did not exhibit a normal distribution. All presented curved are obtained on the testing set.

### 4.1 2L-SPC CONVERGES TO A LOWER PREDICTION ERROR

As a first analysis we report the global prediction error, as computed as the sum of the prediction error ($\epsilon_i$) over all layers (see Fig. 3), and its decomposition among layer (see Fig. 2 and Appendix D fro AT&T database) for different value of sparsity. For scaling reasons, and because the error bars are small we cannot display them on Fig. 2, we thus include them in Appendix Fig. 7. For all the simulations shown in Fig. 2 and Fig. 3, we observed that the global prediction error is lower for the 2L-SPC than for the Hi-La model. As expected, in both models the prediction errors increase similarly when we increase $\lambda_1$ or $\lambda_2$. For all databases and sparsity parameters, Fig. 2 shows that the first layer prediction error of the 2L-SPC is always higher, and the second layer prediction error

**(a)** Distribution of the prediction error among layers when varying $\lambda_1$.

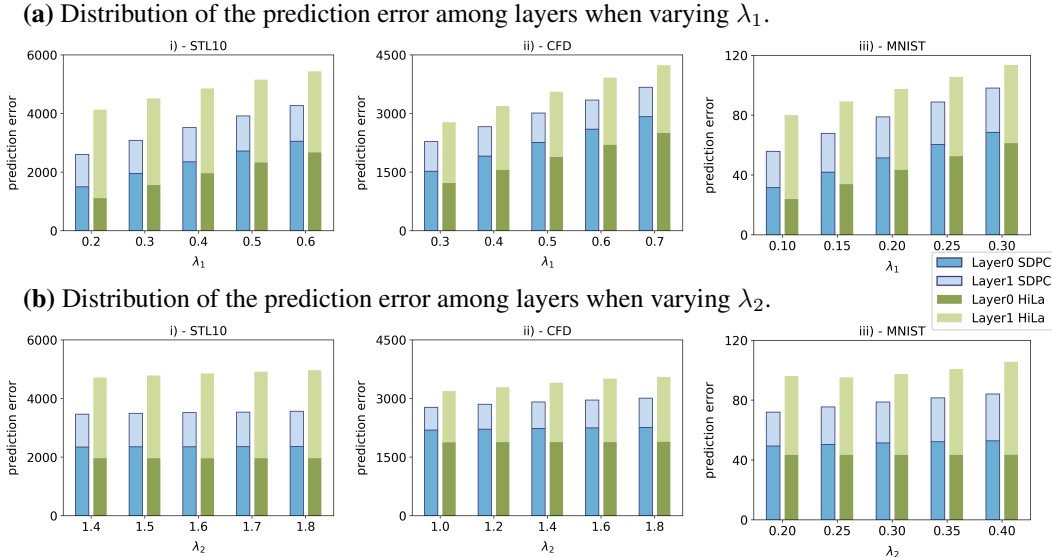

**(b)** Distribution of the prediction error among layers when varying $\lambda_2$.

Figure 2: Evolution of all layers prediction error, evaluated on the testing set, for both 2L-SPC and Hi-La networks and trained on STL-10, CFD and MNIST databases. We vary the first layer sparsity in the top 3 graphs (**a**) and the second layer sparsity in the bottom 3 graphs (**b**).

is always lower than the corresponding Hi-La prediction error. This is expected: while the Hi-La first layer is fully specialized in minimizing the prediction error with the lower level, the 2L-SPC finds a trade-off between lower and higher level prediction errors. In addition, when $\lambda_1$ is increased, the Hi-La first layer prediction error increases faster (+139% for STL-10, +105% for CFD, +157% for MNIST and +93% for AT&T ) than the 2L-SPC first layer prediction error (+103% for STL-10, +90% for CFD, +117% for MNIST and +76% for AT&T). This suggests that a part of the penalty induced by the increase of $\lambda_1$ is absorbed by the second layer in the case of the 2L-SPC. This is supported by the fact that the second layer prediction error of the 2L-SPC is in general increasing slightly faster than the Hi-La second layer error. When $\lambda_2$ is increased, the prediction error of the first layer of the Hi-La model is not varying whereas the 2L-SPC first layer prediction error is slightly increasing (+1% for STL-10, +3% for CFD, +7.1% for MNIST and +3.3% for AT&T). The explanation here is straightforward: while the first-layer loss of the 2L-SPC includes the influence of the upper-layer, the Hi-La doesn't have such a mechanism. It suggests that the inter-layer feedback connection of the 2L-SPC transfers a part of the extra-penalty coming from the increase of $\lambda_2$ in the first layer. Fig.3 i) and ii) show the mapping of the global prediction error when we vary the sparsity of each layer for the 2L-SPC and Hi-La, respectively. These heatmaps confirm what has been observed in Fig.2 and extend it to a larger range of sparsity values: both models' losses are more sensitive to a variation of $\lambda_1$ than to a change in $\lambda_2$. Fig.3 iii) is a heatmap of the relative difference between the 2L-SPC and the Hi-La global losses. It shows that the minimum relative difference between 2L-SPC and Hi-La (10.6%) is reached when $\lambda_1$ is maximal and $\lambda_2$ is minimal, and the maximum relative difference (19.99%) is reached when both $\lambda_1$ and $\lambda_2$ are minimal. It suggests that the previously observed mitigation mechanism originated by the feedback connection is more efficient when the sparsity of the first layer is lower. All these observations point in the same direction: the 2L-SPC framework mitigates the global prediction error thanks to a better distribution of the prediction error among layers. This mechanism is even more pronounced when the sparsity of the first layer is lower. Surprisingly, while the inter-layer feedback connection of the 2L-SPC imposes more constraints on the state variables, it also happens to generate less global prediction error.

### 4.2    2L-SPC HAS A FASTER INFERENCE PROCESS

One may wonder if this lower prediction error is not achieved at the cost of a slower inference process. To address this concern, we report for both models the number of iterations needed by the

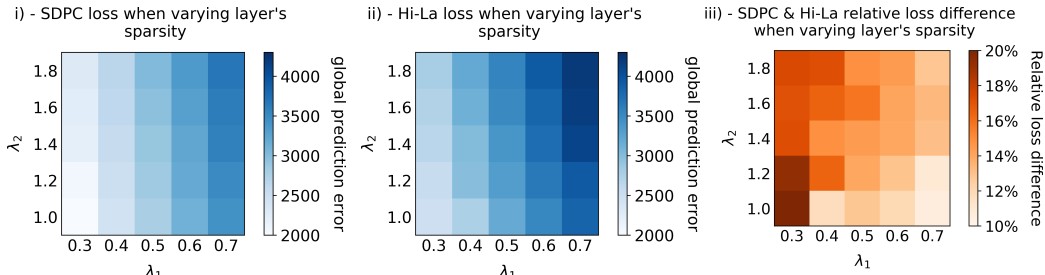

Figure 3: Heatmaps of the global loss error when varying layers' sparsity for 2L-SPC (i) and Hi-La (ii) on CFD database. (iii) shows the heatmap of the relative difference between the Hi-La and the 2L-SPC global losses when varying the layers' sparsity

inference process to converge towards a stable state on the testing set. Fig. 4 shows the evolution of this quantity, for STL-10, CFD and MNIST databases (see Appendix E for AT&T database), when varying both layers' sparsity. For all the simulations, the 2L-SPC needs less iteration than the Hi-La model to converge towards a stable state. We also observe that the data dispersion is, in general, more pronounced for the Hi-La model. In addition to converging to lower prediction error, the 2L-SPC is also decreasing the number of iterations in the inference process to converge towards a stable state.

**(a)** Number of iterations of the inference when varying $\lambda_1$

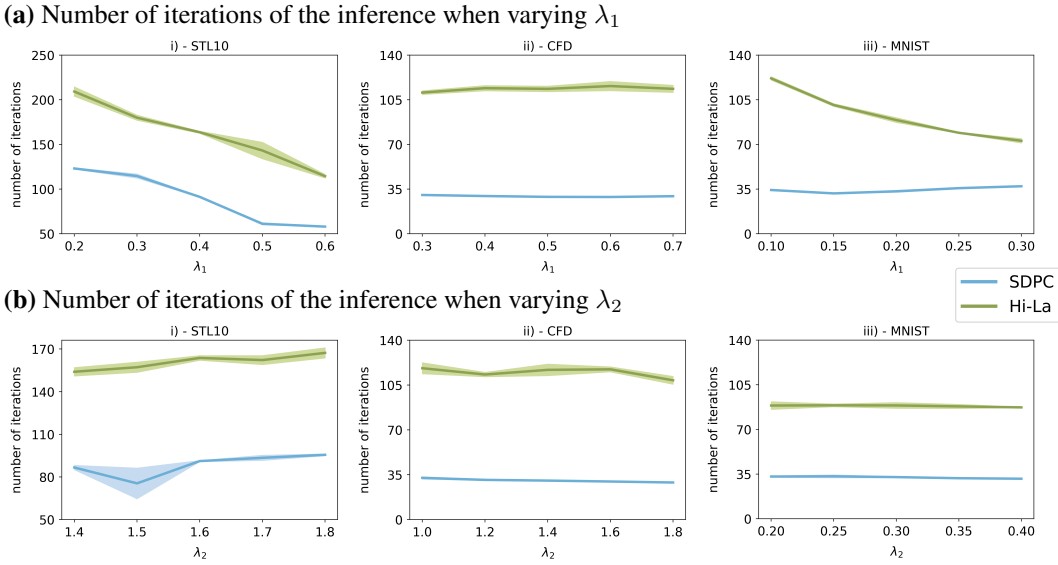

**(b)** Number of iterations of the inference when varying $\lambda_2$

Figure 4: Evolution of the number of iterations needed to reach stability criterium for both 2L-SPC and Hi-La networks on the testing set of STL-10, CFD and MNIST databases. We vary the first layer sparsity in the top 3 graphs (**a**) and the second layer sparsity in the bottom 3 graphs (**b**). Shaded areas correspond to mean absolute deviation on 7 runs. Sometimes the dispersion is so small that it looks like there is no shade.

### 4.3 2L-SPC LEARNS FASTER

Fig. 5 shows the evolution of the global prediction error during the dictionary learning stage and evaluated on the testing set (see Appendix. F for AT&T database). For the all databases, the 2L-SPC model reaches its minimal prediction error before the Hi-La model. The convergence rate of both models is comparable, but the 2L-SPC has a much lower prediction error in the very first epochs.

The inter-layer feedback connection of the 2L-SPC pushes the network towards lower prediction error since the very beginning of the learning.

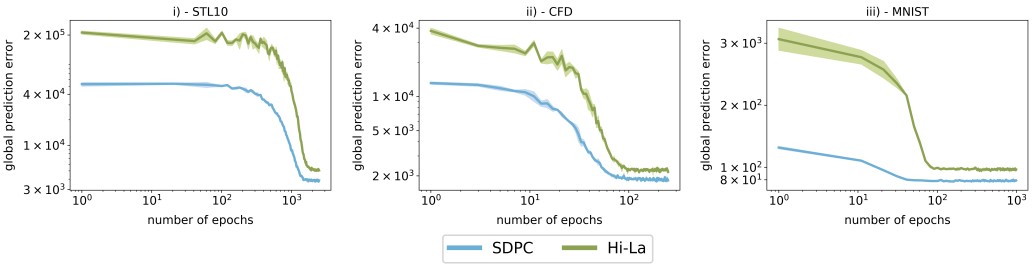

Figure 5: Evolution of the global prediction error during the training evaluated on the STL-10, CFD and MNIST testing sets. Shaded areas correspond to mean absolute deviation on 7 runs. All graphs have a logarithmic scale in both x and y-axis.

## 4.4 QUALITATIVE ANALYSIS OF THE FEATURES

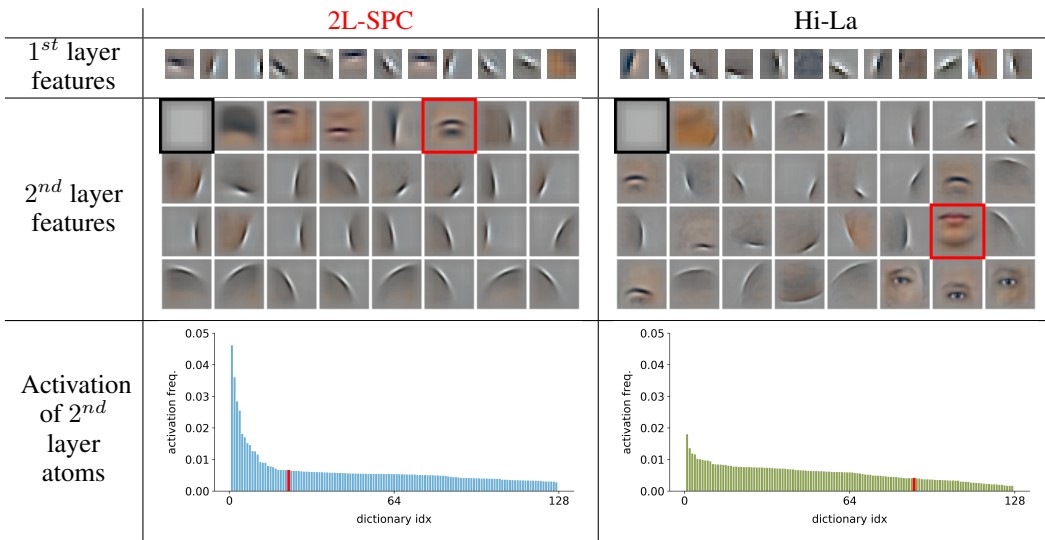

Figure 6: Hi-La and 2L-SPC RFs obtained on the CFD database (with $\lambda_1 = 0.3$, $\lambda_2 = 1.8$) and their associated second layer activation probability histogram. The first and second layer RFs have a size of $9 \times 9$ px and $33 \times 33$ px respectively. For the first layer RFs, we randomly selected 12 out of 64 atoms. For the second layer RFs, we sub-sampled 32 out of 128 atoms ranked by their activation probability in descending order. For readability, we removed the most activated filter (RF framed black) in 2L-SPC and Hi-La second layer activation histogram. The activation probability of the RFs framed in red are shown as a red bar in the corresponding histogram.

Another way to grasp the impact of the inter-layer feedback connection is to visualize its effect on the dictionaries. To make human-readable visualizations of the learned dictionaries, we back-project them into the image space using a cascade of transposed convolution (see Appendix Fig.11). These back-projection are called Receptive Fields (RFs). Fig. 6 shows some of the RFs of the 2 layers and the second layer activation probability histogram for both models when they are trained on the CFD database. In general, first layer RFs are oriented Gabor-like filters, and second layer RFs are more specific and represent more abstract concepts (curvatures, eyes, mouth, nose...). Second layer RFs present longer curvatures in the 2L-SPC than in the Hi-La model: they cover a bigger part of the input image, and include more contextual and informative details. In some extreme cases, the Hi-La second layer RFs are over-fitted to specific faces and do not describe the generality of the

concept of face. The red-framed RFs highlights one of these cases: the corresponding activation probabilities are $0.25\%$ and $0.69\%$ for Hi-La and 2L-SPC respectively. This is supported by the fact that the lowest activation probability of the second layer's atoms is higher for the 2L-SPC than for the Hi-La ($0.30\%$ versus $0.16\%$). This phenomenon is even more striking when we sort all the features by activation probabilities in descending order (see Appendix Figures 13). We filter out the highest activation probability (corresponding to the low-frequency filters highlighted by black square) of both Hi-La and 2L-SPC to keep good readability of the histograms. All the filters are displayed in Appendix Fig. 12, Fig. 13, Fig. 14 and Fig. 15, for STL-10, CFD, MNIST and AT&T RFs respectively. The atoms' activation probability confirm the qualitative analysis of the RFs: the features learned by the 2L-SPC are more generic and informative as they describe a wider range of images.

## 5 CONCLUSION

What are the computational advantages of inter-layer feedback connections in hierarchical sparse coding algorithms? We answered this question by comparing the Hierarchical Lasso (Hi-La) and the 2-Layers Sparse Predictive Coding (2L-SPC) models. Both are identical in every respect, except that the 2L-SPC brings inter-layer feedback connections. This extra-connection forces the internal state variables of the 2L-SPC to converge toward a trade-off between on one hand an accurate prediction passed by the lower-layer and on the other hand a facilitated predictability by the upper-layer. Experimentally, we demonstrated on 4 different databases and for a 2-layered network that the inter-layer feedback top-down connection (i) mitigates the overall prediction error by distributing it among layers, (ii) accelerates the convergence towards a stable internal state and (iii) accelerates the learning process. Besides, we qualitatively observed that top-down connections bring contextual information that helps to extract more informative and less over-fitted features.

The 2L-SPC holds the novelty to consider Hierarchical Sparse Coding as a combination of local sub-problems that are tightly related. This a crucial difference with CNNs that are trained by back-propagating gradients from a global loss. To the best of our knowledge the 2L-SPC is the first one that leverage local sparse coding into a hierarchical and unsupervised algorithms (the ML-CSC from (Sulam et al., 2018) is equivalent to a one layer sparse coding algorithm (Aberdam et al., 2019), and the ML-ISTA from (Sulam et al., 2019) is trained using supervised learning).

Moreover, even if our results are robust as they hold for 4 different databases and with a large spectrum of first and second layer sparsity, further work will be conducted to generalize our results to deeper networks and different sparse coding algorithms such as Coordinate Descent or ADMM. Further studies will show that our 2L-SPC framework could be used for practical applications like image inpainting, denoising, or image super-resolution.

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

# A  2L-SPC PSEUDO CODE

---

**Algorithm 2:** 2L-SPC inference algorithm

---

**input** : image: $\boldsymbol{x}$, dictionaries: $\{\boldsymbol{D}_i\}_{i=1}^L$, $\ell_1$-penalty parameters: $\{\lambda_i\}_{i=1}^L$, stability threshold: $T_{stab}$

$\boldsymbol{\gamma}_0^t = \boldsymbol{x}$
$\{\boldsymbol{\gamma}_i^0\}_{i=1}^L = \boldsymbol{0}, \quad \{\boldsymbol{\gamma}_{m_i}^1\}_{i=1}^L = \boldsymbol{0}$ # Initializing layer state variables and FISTA momentum
$\alpha^1 = 1$ # Initializing momentum strength
$\eta_{c_i} = \dfrac{1}{max(eigen\_value(\boldsymbol{D}_i^T \boldsymbol{D}_i))}$
$Stable = False$ # Initializing the stability criterion
$t = 0$

**while** $Stable == False$ **do**
$\quad t \mathrel{+}= 1$
$\quad \alpha^{t+1} = \dfrac{1 + \sqrt{1 + 4(\alpha^t)^2}}{2}$
$\quad$ **for** $i = 1$ **to** $L$ **do**
$\quad\quad$ # Update lower-layer error
$\quad\quad \boldsymbol{\epsilon}_{LL} = \boldsymbol{\gamma}_{m_{i-1}}^t - \boldsymbol{D}_i^T \boldsymbol{\gamma}_{m_i}^t$
$\quad\quad$ # Update the upper-layer error
$\quad\quad$ **if** $i \neq L$ **then**
$\quad\quad\quad \boldsymbol{\epsilon}_{UL} = \boldsymbol{\gamma}_{m_i}^t - \boldsymbol{D}_{i+1}^T \boldsymbol{\gamma}_{m_{i+1}}^t$
$\quad\quad$ **else**
$\quad\quad\quad \boldsymbol{\epsilon}_{UL} = \boldsymbol{0}$
$\quad\quad \boldsymbol{\gamma}_i^t = \mathcal{T}_{\eta_{c_i}\lambda_i}\big(\boldsymbol{\gamma}_{m_i}^t + \eta_{c_i}\boldsymbol{D}_i\boldsymbol{\epsilon}_{LL} - \eta_{c_i}\boldsymbol{\epsilon}_{UL}\big)$ # Update layer state variables
$\quad\quad \boldsymbol{\gamma}_{m_i}^{t+1} = \mathcal{T}_0\Big(\boldsymbol{\gamma}_i^t + \big(\dfrac{\alpha^t - 1}{\alpha^{t+1}}\big)\big(\boldsymbol{\gamma}_i^t - \boldsymbol{\gamma}_i^{t-1}\big)\Big)$ # Update the FISTA momentum
$\quad$ **if** $\displaystyle\bigwedge_{i=1}^L \Big(\dfrac{\|\boldsymbol{\gamma}_i^t - \boldsymbol{\gamma}_i^{t-1}\|_2}{\|\boldsymbol{\gamma}_i^t\|_2} < T_{stab}\Big)$ **then**
$\quad\quad Stable = True$ # Update the stability criterion

**return** $\{\boldsymbol{\gamma}_i^t\}_{i=1}^L$

---

**Note:** $\mathcal{T}_\alpha(\cdot)$ denotes the element-wise non-negative soft-thresholding operator. A fortiori, $\mathcal{T}_0(\cdot)$ is a rectified linear unit operator. # comments are comments.

# B EVOLUTION OF THE GLOBAL PREDICTION ERROR WITH ERROR BAR

**(a)** Global prediction error when varying $\lambda_1$.

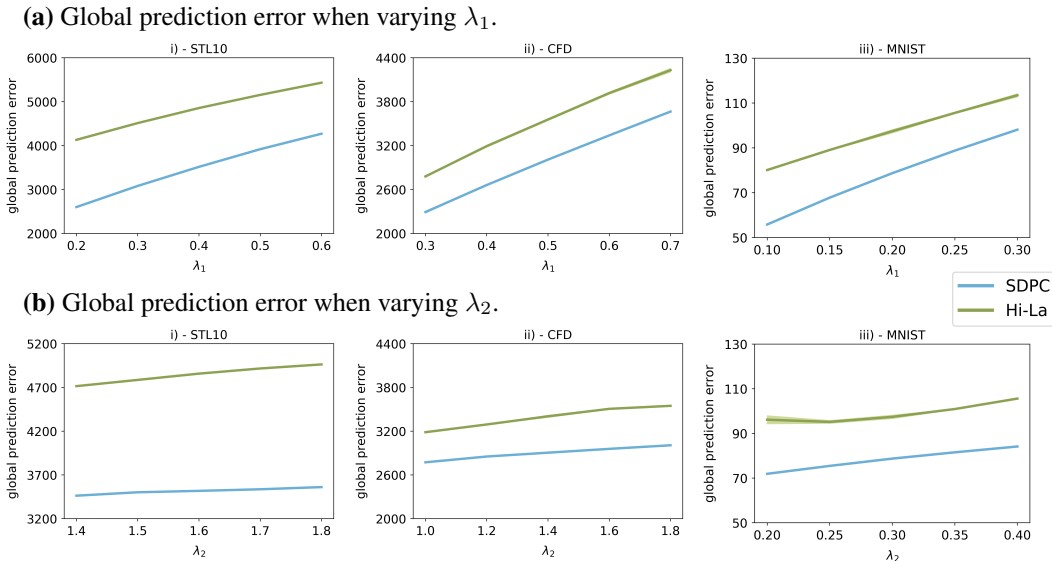

**(b)** Global prediction error when varying $\lambda_2$.

Figure 7: Evolution of the global prediction error evaluated on the testing set for both 2L-SPC and Hi-La networks. We vary the first layer sparsity in the top 3 graphs (**a**) and the second layer sparsity in the bottom 3 graphs (**b**). Experiments have been conducted on STL-10, CFD and MNIST-databases. Shaded areas correspond to mean absolute deviation on 7 runs. Sometimes the dispersion is so small that it looks like there is no shade.

# C 2L-SPC PARAMETERS ON ATT

Table 2: Network architectures, training and simulation parameters on AT&T database. The size of the convolutional kernels are shown in the format: [# features, # channels, width, height] (stride). To describe the range of explored parameters during simulations, we use the format $[0.3 : 0.7 :: 0.1, 0.5]$ ,which means that we vary $\lambda_1$ from 0.3 to 0.7 by step of 0.1 while $\lambda_2$ is fixed to 0.5.

|  |  | ATT Database |
|---|---|---|
| network param. | $D_1$ size | [64, 1, 9, 9] (3) |
|  | $D_2$ size | [128, 64, 9, 9] (1) |
|  | $T_{stab}$ | 5e-4 |
| training param. | # epochs | 1000 |
|  | $\eta_{L_1}$ | 1e-4 |
|  | $\eta_{L_2}$ | 5e-3 |
| simu. param. | $\lambda_1$ range | $[0.3 : 0.7 :: 0.1, 1]$ |
|  | $\lambda_2$ range | $[0.5, 0.6 : 1.6 :: 0.2]$ |

# D  PREDICTION ERRORS DISTRIBUTION ON AT&T

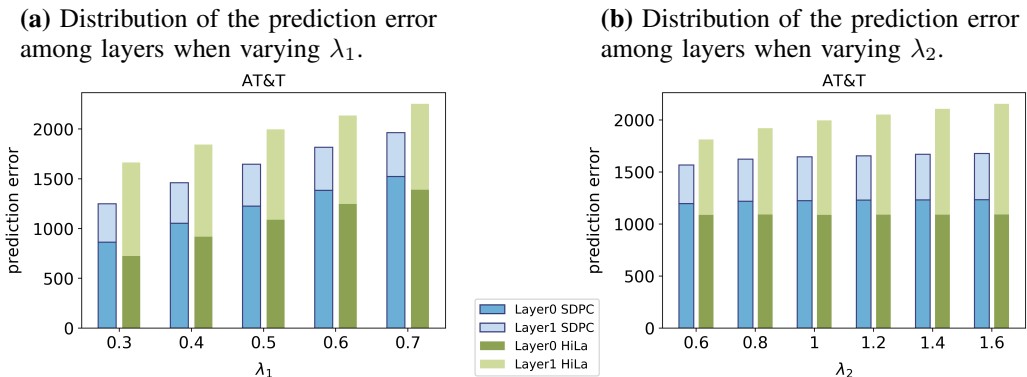

**(a)** Distribution of the prediction error among layers when varying $\lambda_1$.

**(b)** Distribution of the prediction error among layers when varying $\lambda_2$.

Figure 8: Evolution of all layers prediction error, evaluated on the testing set, for both 2L-SPC and Hi-La networks and trained on AT&T. We vary the first layer sparsity in (**a**) and the second layer sparsity in (**b**).

# E  NUMBER OF ITERATION OF THE INFERENCE ON AT&T

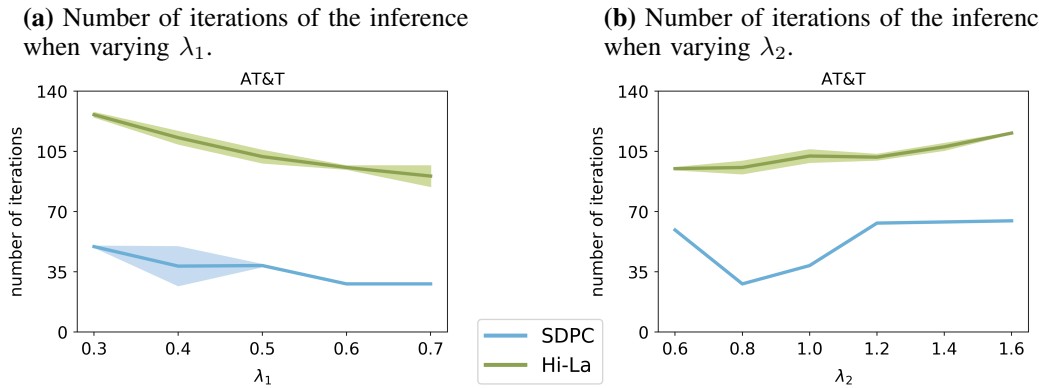

**(a)** Number of iterations of the inference when varying $\lambda_1$.

**(b)** Number of iterations of the inference when varying $\lambda_2$.

Figure 9: Evolution of the number of iterations needed to reach stability criterium for both 2L-SPC and Hi-La networks on the AT&T testing set. We vary the first layer sparsity in (**a**) and the second layer sparsity in (**b**). Shaded areas correspond to mean absolute deviation on 7 runs. Sometimes the dispersion is so small that it looks like there is no shade.

# F EVOLUTION OF PREDICTION ERROR DURING TRAINING

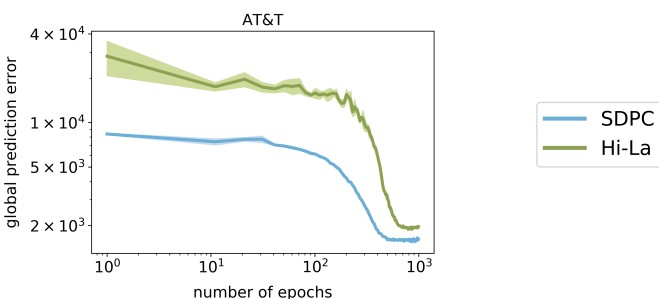

Figure 10: Evolution of the global prediction error during the training for the ATT testing set. Shaded areas correspond to mean absolute deviation on 7 runs. The graph have a logarithmic scale in both x and y-axis.

# G ILLUSTRATION OF THE BACK-PROJECTION MECHANISM

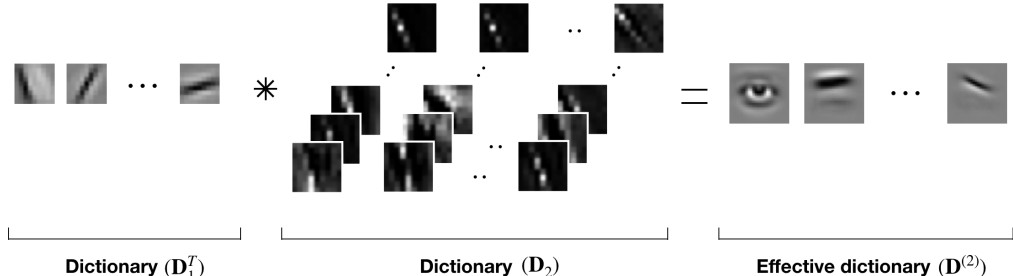

Figure 11: Generation of the second-layer effective dictionary. The result of this back-projection is called effective dictionary and could be assimilate to the notion of preferred stimulus in neuroscience. In a general case, the effective dictionary at layer $i$ is computed as follow: $\mathbf{D}_i^{\text{eff},T} = \mathbf{D}_0^T..\mathbf{D}_{i-1}^T\mathbf{D}_i^T$ (Sulam et al., 2018).

# H   2L-SPC AND HI-LA RFS ON STL10

**(a)** 2L-SPC first layer RFs

**(b)** Hi-La first layer RFs

**(c)** 2L-SPC second layer RFs

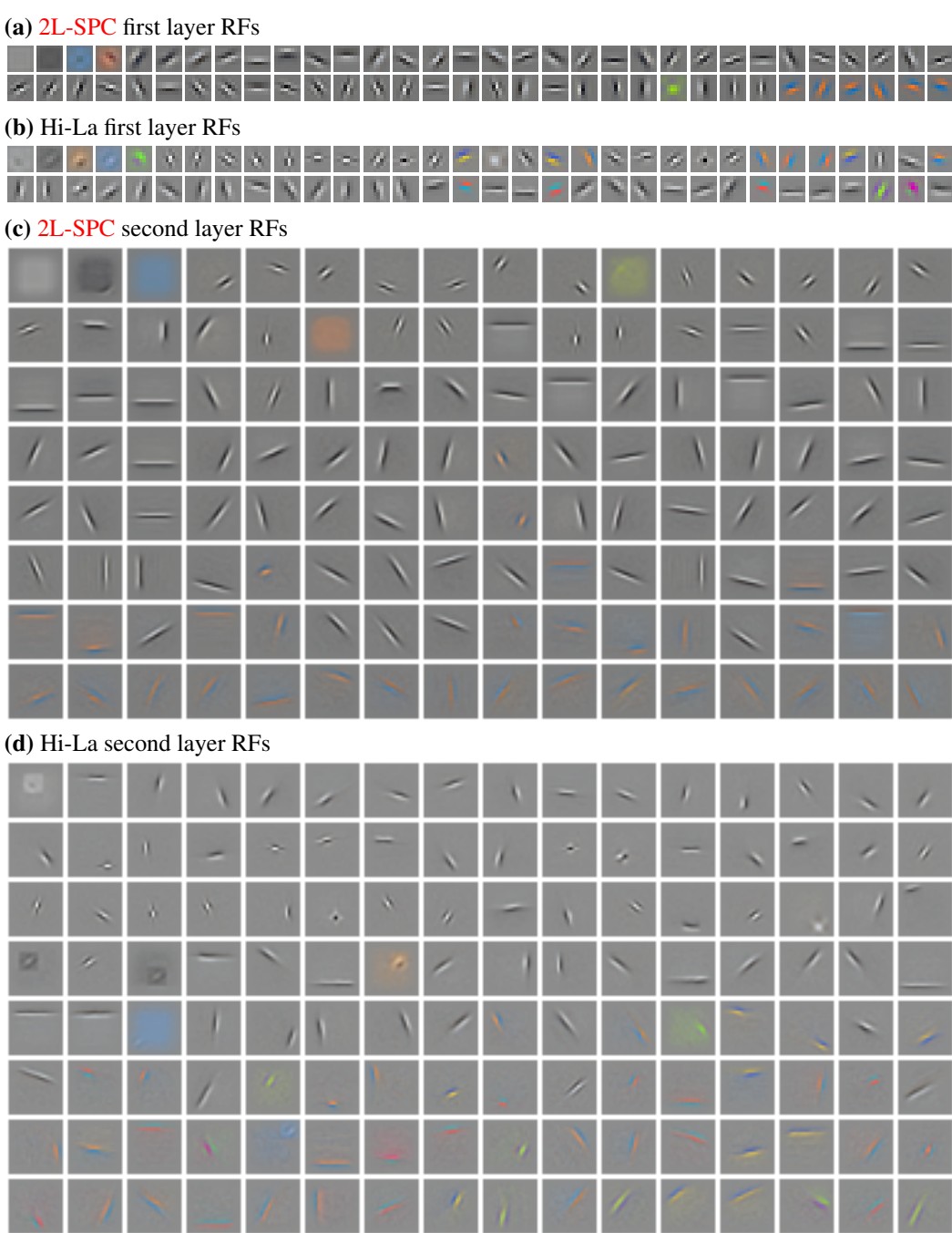

**(d)** Hi-La second layer RFs

Figure 12: 2L-SPC (**a** & **c**) and Hi-La (**b** & **d**) effective dictionaries obtained on the STL-10 database, with sparsity parameter: ($\lambda_1$=0.5,$\lambda_2$=1). All other parameters are those described in Table 1 for the STL-10 database. Atoms are sorted by activation probabilities in a descending order. The visualization shown here is the projection of the dictionaries into the input space. First layer effective dictionaries have a size of $8 \times 8$ px (**a** & **b**) and second layer RFs have a size of $22 \times 22$ (**c** & **d**) px respectively.

# I  2L-SPC AND HI-LA RFS ON CFD

**(a)** 2L-SPC first layer RFs

**(b)** Hi-La first layer RFs

**(c)** 2L-SPC second layer RFs

**(d)** Hi-La second layer RFs

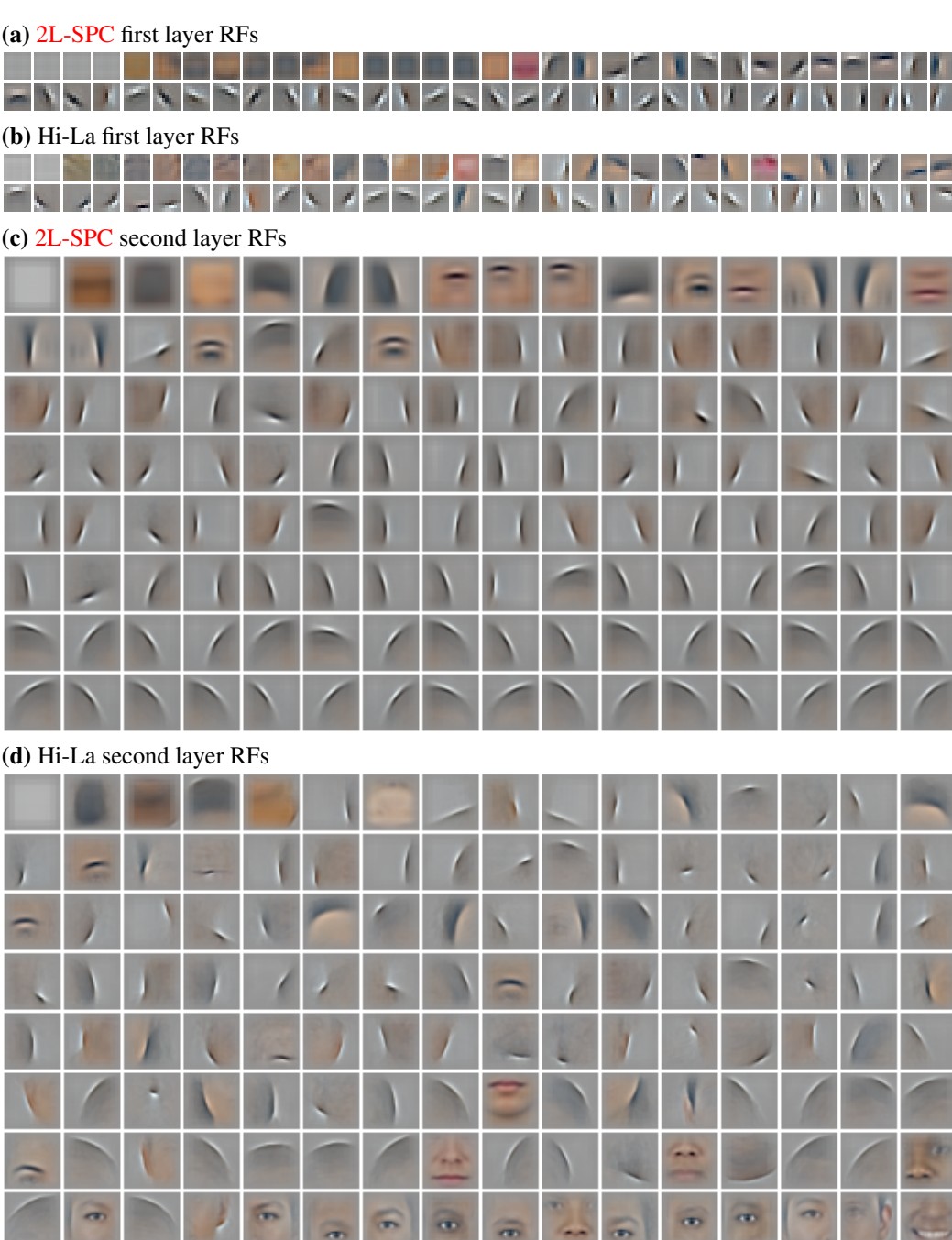

Figure 13: 2L-SPC (**a** & **c**) and Hi-La (**b** & **d**) effective dictionaries obtained on the CFD database, with sparsity parameter: ($\lambda_1$=0.3,$\lambda_2$=1.8). All other parameters are those described in Table 1 for the CFD database. Atoms are sorted by activation probabilities in a descending order. The visualization shown here is the projection of the dictionaries into the input space. First layer effective dictionaries have a size of $9 \times 9$ px (**a** & **b**) and second layer RFs have a size of $33 \times 33$ (**c** & **d**) px respectively.

## J  2L-SPC AND HI-LA RFS ON MNIST

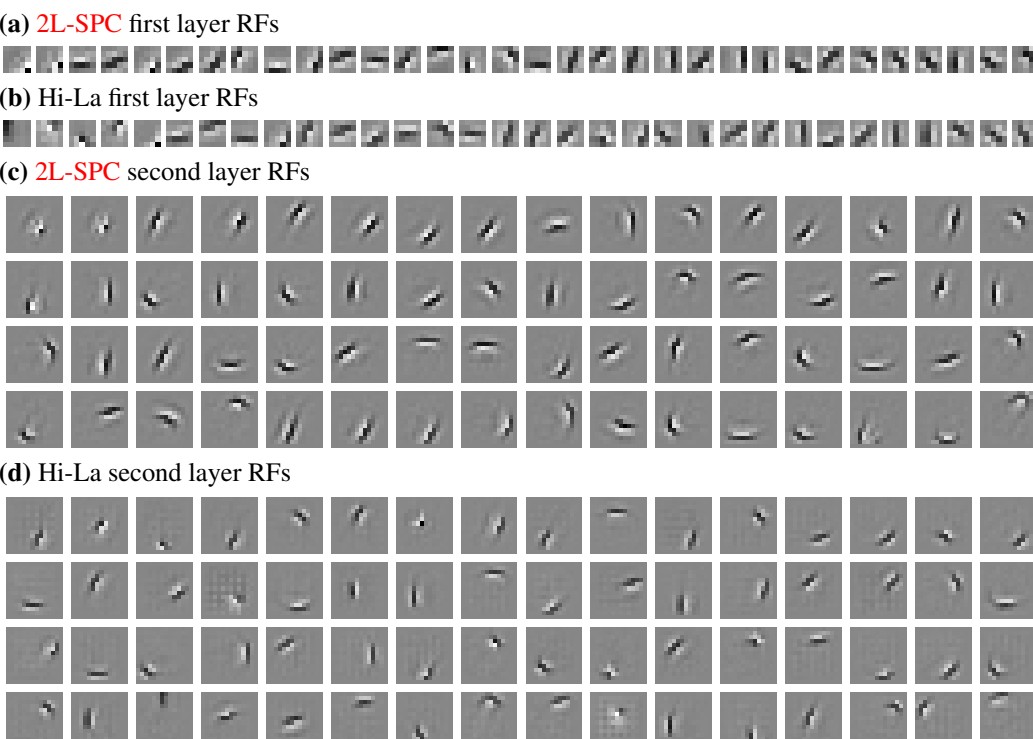

**(a)** 2L-SPC first layer RFs

**(b)** Hi-La first layer RFs

**(c)** 2L-SPC second layer RFs

**(d)** Hi-La second layer RFs

Figure 14: 2L-SPC (**a** & **c**) and Hi-La (**b** & **d**) effective dictionaries obtained on the MNIST database, with sparsity parameter: ($\lambda_1$=0.2,$\lambda_2$=0.3). Atoms are sorted by activation probabilities in a descending order. All other parameters are those described in Table 1 for the MNIST database. The visualization shown here is the projection of the dictionaries into the input space. First layer effective dictionaries have a size of $5 \times 5$ px (**a** & **b**) and second layer RFs have a size of $14 \times 14$ (**c** & **d**) px respectively.

# K  2L-SPC AND HI-LA RFS ON AT&T

**(a)** 2L-SPC first layer RFs

**(b)** Hi-La first layer RFs

**(c)** 2L-SPC second layer RFs

**(d)** Hi-La second layer RFs

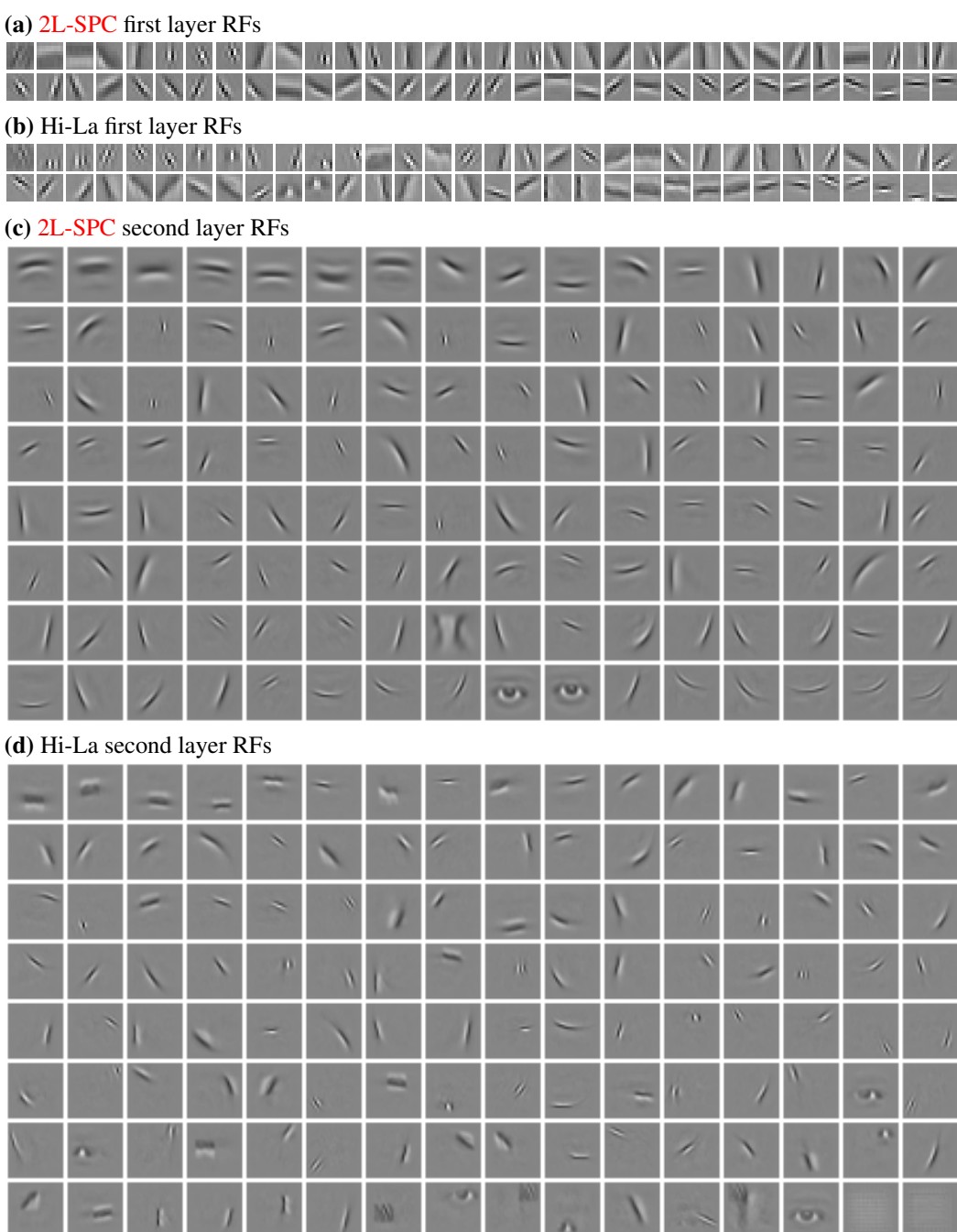

Figure 15: 2L-SPC (**a** & **c**) and Hi-La (**b** & **d**) effective dictionaries obtained on the AT&T database, with sparsity parameter: ($\lambda_1$=0.5,$\lambda_2$=1). All other parameters are those described in Table 2 for the AT&T database. Atoms are sorted by activation probabilities in a descending order. The visualization shown here is the projection of the dictionaries into the input space. First layer effective dictionaries have a size of $9 \times 9$ px (**a** & **b**) and second layer RFs have a size of $26 \times 26$ (**c** & **d**) px respectively.

