# OpenReview forum: "Effect of top-down connections in Hierarchical Sparse Coding"
_ICLR.cc/2020/Conference — Reject_

### Official Review · AnonReviewer2 · 2019-10-23
**Official Blind Review #2**

**Rating:** 3

**Review:**

This paper proposes Sparse Deep Predictive Coding (SDPC) to access the impact of the inter layer feedback, which is suggested by neuro-scientific evidence. The SDPC model is compared with HILA on 2 different databases, and the experimental results show that SDPC achieved lower prediction error, faster converge rate.

Although the paper is well-motivated and well-written, the potential impact of this work is limited in the community. Moreover, the experiments are not sufficient to valid the advantage of this model. For example, there are only 3 dataset adopted in the experiments and the all these three datasets are quite small. Moreover, there is only one baseline, which seems too weak to show the advantage of the model.

Based on the above points, I tend to reject the paper.

**updated comments**
Thanks for the response and I really appreciate the hard work during the rebuttal. However, the STL-10 is still a small scale dataset, and the baselines provided in the rebuttal are still limited. I will not change the original score.

**Experience Assessment:**

I have published one or two papers in this area.

**Review Assessment: Checking Correctness Of Derivations And Theory:**

I assessed the sensibility of the derivations and theory.

**Review Assessment: Checking Correctness Of Experiments:**

I assessed the sensibility of the experiments.

**Review Assessment: Thoroughness In Paper Reading:**

I read the paper at least twice and used my best judgement in assessing the paper.

---

> ### Author Response · Authors · 2019-11-08
> **Response to reviewer #2**
>
> We thank the reviewer #2 for pointing out the small-size of databases. We circumvent this issue by running new simulations on the STL-10 database (100 000 images divided into a training (90 000 images) and a testing set (10000 images). This simulation is in the updated version of the paper. Note that the additional runs on STL-10 needed to compute the error-bar are still running. As soon as the simulations will be done, we will update the paper accordingly. To keep the paper concise and clear, we replaced the previous experiment an AT&T database (the smaller dataset) by those on STL-10. We put the ‘old’ simulation an ATT in the supplementary information section. We hope this major update will convince that reviewer that our paper can attract the interest of a broad audience.
>
> Reviewer #2 is absolutely right to raise the lack of baseline. A good baseline will consist in an unsupervised hierarchical sparse coding model that has the same number of parameters than the models presented in the paper. This last constraint prevent us to us sparse auto-encoder (which has twice the number of parameters because weights of encoder and not tight to those of the decoder). To the best of our knowledge, the only models that are similar (in terms of number of parameters) are those from the Michael Elad team: The ML-CSC [1] and the ML-ISTA [2]. However, both models shown significant enough difference to make the comparison with the SDPC not relevant. First, the Elad team itself has demonstrated that the ML-CSC was equivalent to a single layer network that was too constrained to leverage hierarchical information [3]. In addition, the inference process of the ML-CSC does not explicitly computes intermediate state-variables. These two points make the comparison with the SDPC irrelevant. Second, the ML-ISTA could have been a good comparison point if the dictionaries were not learned with supervised training. The ML-ISTA uses a back-propagation over of cross-entropy loss function as a training mechanism. However, the reviewer rightly pointed out the fact that we did not justify the absence of baseline. We thus updated the article to highlight the specificity of the SDPC that makes it complicated to compare to other state-of-the-art algorithms.
>
> [1] Sulam, Jeremias, et al. "Multilayer convolutional sparse modeling: Pursuit and dictionary learning." IEEE Transactions on Signal Processing 66.15 (2018): 4090-4104.
>
> [2] Sulam, Jeremias, et al. "On multi-layer basis pursuit, efficient algorithms and convolutional neural networks." IEEE transactions on pattern analysis and machine intelligence (2019).
>
> [3] Aberdam, Aviad, Jeremias Sulam, and Michael Elad. "Multi-Layer Sparse Coding: The Holistic Way." SIAM Journal on Mathematics of Data Science 1.1 (2019): 46-77.

---

### Official Review · AnonReviewer1 · 2019-10-23
**Official Blind Review #1**

**Rating:** 3

**Review:**

This paper presents a study that compares two techniques for Hierarchical Sparse Coding. This is the problem of learning sparse representations in multi-layer (but not necessarily deep) models. The first method applies Lasso at each layer in a bottom up fashion, while the second method — introduced in the submitted work — adds an additional term which propagates information in a top-down manner. It is found that the top-down term is beneficial in terms of reducing predictive error and can learn faster. In terms of novelty the new term, the paper does not make a breakthrough contribution, but I consider this to be sufficient. Moreover, the paper is well written and presents some interesting results. However I am not entirely sure that the impact of these results will attract the interest of a broad audience since the experiments presented are on “small datasets” and with some rather shallow neural nets.

**Experience Assessment:**

I have read many papers in this area.

**Review Assessment: Checking Correctness Of Derivations And Theory:**

N/A

**Review Assessment: Checking Correctness Of Experiments:**

I assessed the sensibility of the experiments.

**Review Assessment: Thoroughness In Paper Reading:**

I made a quick assessment of this paper.

---

> ### Author Response · Authors · 2019-11-08
> **Response to reviewer #1**
>
> We thank the reviewer #1 for pointing out the small-size of databases. We circumvent this issue by running new simulations on the STL-10 database (100 000 images divided into a training and a testing set). This simulation is in the updated version of the paper. Note that the additional runs on STL-10 needed to compute the error-bar are still running. As soon as the simulations will we done, we will update the paper accordingly. One can observe that the results hold for bigger databases like STL-10.  To keep the paper concise and clear, we replaced the previous experiment on AT&T database (the smaller dataset) by those on STL-10. We put the ‘old’ simulation an ATT in the supplementary information section. We hope this major update will convince the reviewer that our paper can attract the interest of a broad audience.
>
> The reviewer #1 correctly raised the fact that we are describing a 2-layer network here, whereas other neural network (mostly CNN trained with back-prop) are deeper. We first propose to rename our algorithm: 2-Layers Sparse Predictive Coding (2L-SPC) instead of Sparse Deep Predictive Coding (SDPC) to fit the claim of this paper. Second, we updated our conclusion section to highlight the novelty of the proposed training method. The SDPC uses local update rules that first needs to be investigated in smaller network before including them in deep architecture. Focusing on this 2-layer network with and without feedback allows us to observe the performance of this new structure applicable for deeper networks. Indeed, we are now working on deeper architectures, and it would be the subject of subsequent paper. We propose to better highlight this point in the conclusion

---

### Official Review · AnonReviewer3 · 2019-10-24
**Official Blind Review #3**

**Rating:** 3

**Review:**

This work proposed a new model called Sparse Deep Predictive Coding, which introduced top-down connections between consecutive layers, to improve the solutions to hierarchical sparse coding (HSC) problems. Instead of decomposing the HSC problem into independent subproblems, the proposed model added a new term to the loss function, which represents the influence of the latter-layer on the current layer.

#Pros:
-- The proposed model adopted the idea from predictive coding and came up with a relatively novel idea for HSC problems.
-- The experiments are solid. The experiments evaluated the proposed methods with different hyper-parameter settings and three real-world datasets.
-- The figures in the result sections are well designed and concise.

#Cons:
-- The mathematical description of the main problem and the proposed model is not clear. For example, the dimensionality for the variables in Eq.(1) is not clarified.
-- The test procedure is not clear. How the internal state variables are obtained for the test set is not clarified.
-- The proposed model was only compared with a basic Hierarchical Lasso network. There are not any state-of-art methods included as baseline methods.

#Detailed comments:

(1) The proposed model is named as sparse DEEP predictive coding, however, the experiments only considered the SDPC and Hi-La networks with 2 layers. I am wondering if a deeper structure will improve the performance?

(2) For the structure shown in Fig.1, the decoding dictionaries are $D^T_i$, but I am confused why the encoding dictionaries are reciprocal to encoding dictionaries. Does it come from the optimization updates shown in Eq.(3)?

(3) According to Eq.(1), $x$ is a vector and $D$ is a 2d matrix. However, the real inputs in the experiments are images and $D$ is a convolutional filter with 4 dimensions. How are the matrices reshaped?

(4) For section 2.2 and 2.3, the number/index of samples is not shown in the loss function for training. The loss should be over the whole training set. Besides, the test procedure is not clarified.

(4) The number of iterations using FISTA of SDPC and Hi-La networks is shown to compare the rate of convergence. However, considering both models are solving a lasso-type regression problem, I would suggest using coordinate descent for optimization.

(5) For the main result of prediction error, why is the “global prediction error” more important than the reconstruction error? Is the first-layer prediction error the reconstruction error? If yes, Fig.2 shows that Hi-La has a lower prediction error compared to SDPC for the first layer.

(6) Two minor comments on writing:
(a) It would be better to have a separate section for 2.5 since it describes the dataset and is not related to the proposed model.
(b) A typo of “neuronal implementation” exists in the introduction section.


**Experience Assessment:**

I have read many papers in this area.

**Review Assessment: Checking Correctness Of Derivations And Theory:**

I carefully checked the derivations and theory.

**Review Assessment: Checking Correctness Of Experiments:**

I carefully checked the experiments.

**Review Assessment: Thoroughness In Paper Reading:**

I read the paper thoroughly.

---

> ### Author Response · Authors · 2019-11-08
> **Response to reviewer#3**
>
> We thank reviewer #3 for the detailed review, the valuable feedback and encouraging words.
>
> Concerning the Cons :
> -- As suggested by reviewer #3, we updated the article with more accurate mathematical notations.
>
> -- The training and testing proceesdures has been clarified. Note that the state variables are obtained through an inference process that is exactly the same for testing and training. The only difference between training and testing is that the dictionary learning step occurs only during the training (and not during testing). We updated the paper to clarify this crucial point of the paper, and we thank reviewer #3 for raising points that need clarification.
>
> -- The reviewer #3 is absolutely right to raise the lack of baseline. A good baseline will consist in an unsupervised hierarchical sparse coding model that has the same number of parameters than the models presented in the paper. This last constraint prevents us to use a sparse auto-encoder (which has twice the number of parameters because weights of encoder are not tied to those of the decoder). To the best of our knowledge, the only models that are similar (in terms of number of parameters) are those from the Michael Elad team: The ML-CSC [1] and the ML-ISTA [2]. First, the ML-CSC was shown to be equivalent to a single layer network that was too constrained to leverage hierarchical information [3]. Second, the ML-ISTA could have been a good comparison point if the dictionaries were not learned using supervised training. However, the reviewer rightly pointed out the fact that we did not justify the absence of baseline. We thus updated the article to highlight the specificity of the SDPC that makes it complicated to compare to other state-of-the-art algorithms (see conclusion section).
>
> Concerning detailled comments :
> (1) - The reviewer #3 correctly raised the fact that we are calling ‘deep’ an architecture that has only 2-layers. We propose to renamed our algorithm: 2-Layers Sparse Predictive Coding (2L-SPC) instead of Sparse Deep Predictive Coding (SDPC) to fit the claim of this paper. Focusing on this 2-layer network with and without feedback allows us to observe the performance of this new structure applicable for deeper networks. Indeed, we are now working on deeper architectures, and it would be the subjects of subsequent paper. We propose to better highlight this point in the conclusion
>
> (2) - The reciprocal dictionary is indeed coming from the inference process described in Eq.(3). To remove any ambiguity, we updated the section 2.2 to explicit the origin of the reciprocal dictionary.
>
> (3) - This comment has been clarified using more accurate notation (see section). Note that transforming convolution on 4d tensors into a matrix/vector multiplication requires more than a reshaping. One needs to transform the 4d tensor into a 2d Toeplitz matrix (see introduction of [1], where this transformation is well described). Nevertheless, we have updated the paper to better explained this point (see section 2.1).
>
>
> (4) - We added Eq.6 to clarify the learning loss.
>
> (4) - The reviewer #3 is absolutely right to suggest the use of Coordinate Descent to optimize our Loss function. However, in a convolutional case in which the derivatives of the loss are easily computable, it has been demonstrated that FISTA algorithm is converging faster than coordinate descent [4]. We updated section 2.2 to better motivate our choice for FISTA.
>
> (5) - We thank the reviewer for raising the interesting difference between reconstruction error and global prediction error (i.e. the sum of all local prediction error). In a 2 layer setting, the reconstruction error is equal to norm2(x - D1^{T}.D2^{T}.gamma_{2}). Interestingly, one can easily demonstrate that this reconstruction error is upper-bounded by the global prediction error (using Eq. 1 and then applying triangular inequality). So both notions are tightly linked, and a lower global prediction error implies a lower reconstruction error (the opposite is not necessarily true).
>
> (6) -  We updated the paper according to the reviewer #3 suggestions
>
> —- References
> [1] Sulam, Jeremias, et al. "Multilayer convolutional sparse modeling: Pursuit and dictionary learning." IEEE Transactions on Signal Processing 66.15 (2018): 4090-4104.
>
> [2] Sulam, Jeremias, et al. "On multi-layer basis pursuit, efficient algorithms and convolutional neural networks." IEEE transactions on pattern analysis and machine intelligence (2019).
>
> [3] Aberdam, Aviad, Jeremias Sulam, and Michael Elad. "Multi-Layer Sparse Coding: The Holistic Way." SIAM Journal on Mathematics of Data Science 1.1 (2019): 46-77.
>
> [4] Chalasani, Rakesh, Jose C. Principe, and Naveen Ramakrishnan. "A fast proximal method for convolutional sparse coding." The 2013 International Joint Conference on Neural Networks (IJCNN). IEEE, 2013.

---

### Author Response · Authors · 2019-11-12
**Updated paper as requested by reviewer #1 and reviewer #2**

Now that all the simulations on the new database (STL-10) are done, we release the updated version of the article.

To facilitate the reviewers' work, all the updates are written in red.

We thank the reviewers for suggesting a bigger database. It improves the quality of the paper and should attract the interest of a broader audience. This additional database strengthens our results as one can draw the same conclusion than previously: the SPDC is faster to converge, and converges to lower prediction errors.

---

### Decision · Program_Chairs · 2019-12-19

**Decision:**

Reject

**Comment:**

This paper introduces a new architecture for sparse coding.

The reviewers gave long and constructive feedback that the authors in turn responded at length on. There is consensus among the reviewers that despite contributions this paper in its current form is not ready for acceptance.

Rejection is therefore recommended with encouragement to make updated version for next conference.